# Modelling the epidemiology of residual *Plasmodium vivax* malaria in a heterogeneous host population: A case study in the Amazon Basin

**Rodrigo M. Corder**[1]*, **Marcelo U. Ferreira**[1], **M. Gabriela M. Gomes**[2,3]*

**1** Department of Parasitology, Institute of Biomedical Sciences, University of São Paulo, São Paulo, Brazil,
**2** Liverpool School of Tropical Medicine, Liverpool, United Kingdom, **3** CIBIO-InBIO, Centro de Investigação em Biodiversidade e Recursos Genéticos, and CMUP, Centro de Matemática da Universidade do Porto, Porto, Portugal

☯ These authors contributed equally to this work.
* rodrigo.corder@usp.br (RMC); Gabriela.Gomes@lstmed.ac.uk (MGMG)

**Data Availability Statement:** Because original data files contain information that may potentially lead to the identification of study participants, requests

## Abstract

The overall malaria burden in the Americas has decreased dramatically over the past two decades, but residual transmission pockets persist across the Amazon Basin, where *Plasmodium vivax* is the predominant infecting species. Current elimination efforts require a better quantitative understanding of malaria transmission dynamics for planning, monitoring, and evaluating interventions at the community level. This can be achieved with mathematical models that properly account for risk heterogeneity in communities approaching elimination, where few individuals disproportionately contribute to overall malaria prevalence, morbidity, and onwards transmission. Here we analyse demographic information combined with routinely collected malaria morbidity data from the town of Mâncio Lima, the main urban transmission hotspot of Brazil. We estimate the proportion of high-risk subjects in the host population by fitting compartmental susceptible-infected-susceptible (SIS) transmission models simultaneously to age-stratified vivax malaria incidence densities and the frequency distribution of *P. vivax* malaria attacks experienced by each individual over 12 months. Simulations with the best-fitting SIS model indicate that 20% of the hosts contribute 86% of the overall vivax malaria burden. Despite the low overall force of infection typically found in the Amazon, about one order of magnitude lower than that in rural Africa, high-risk individuals gradually develop clinical immunity following repeated infections and eventually constitute a substantial infectious reservoir comprised of asymptomatic parasite carriers that is overlooked by routine surveillance but likely fuels onwards malaria transmission. High-risk individuals therefore represent a priority target for more intensive and effective interventions that may not be readily delivered to the entire community.

for access to original data should be submitted to the Institutional Review Board of the Institute of Biomedical Sciences, University of São Paulo, Brazil, at cep@icb.usp.br. The study identification number is CEPSH 1368.

**Funding:** This work was supported by the Fundação de Amparo à Pesquisa do Estado de São Paulo (FAPESP; http://www.fapesp.br/en/), Brazil (grant 2016/18740-9) and by the National Institute of Allergy and Infectious Diseases, National Institutes of Health (https://www.niaid.nih.gov/), United States of America (grant U19 AI089681). RMC receives a doctoral fellowship from the Conselho Nacional de Desenvolvimento Científico e Tecnológico (CNPq; http://cnpq.br/), which also provides a senior research scholarship to MUF. MGMG received funding from the Fundação para a Ciência e a Tecnologia (https://www.fct.pt/index.phtml.en), Portugal (grant IF/01346/2014).The funders had no role in study design, data collection and analysis, decision to publish, or preparation of the manuscript.

**Competing interests:** The authors have declared that no competing interests exist.

## Author summary

Malaria transmission models that disregard risk heterogeneity at the community level, classifying individuals as uniformly susceptible or infected, may not properly recapitulate the epidemiology of malaria in real-life settings. Here we fit a compartmental susceptible-infected-susceptible model to malaria morbidity data from Mâncio Lima, the main urban transmission hotspot of Brazil, and estimate that 20% of the urban residents contribute 86% of the overall vivax malaria burden in the town. Despite the low average force of infection, one order of magnitude lower that in rural Africa, high-risk individuals experience enough repeated infections to develop clinical immunity and constitute an asymptomatic reservoir that fuels onwards malaria transmission. Therefore, these high-risk subjects account for the paradoxical finding of clinical immunity and frequent asymptomatic parasite carriage in low-endemicity Amazonian communities. We argue that mathematical models accounting for risk heterogeneity are crucial to plan and evaluate malaria control and elimination interventions targeted to high-risk groups in communities, municipalities, and regions.

## Introduction

Heterogeneity in the risk of infection with several pathogens has been repeatedly documented in human populations, with 20% of the hosts typically harbouring 80% of the pathogen burden in the community [1]. For example, residents in the same village in rural Africa may greatly differ in their malaria risk, leading to over-dispersed frequency distributions of malaria attacks per person over time, with few subjects in the community experiencing frequent infection and disease [2].

One source of malaria risk heterogeneity is the varying hosts' exposure to the pathogen, which can be measured as the number of infectious mosquito bites per host per unit of time, termed the entomological inoculation rate (EIR). About 20% of the children are estimated to receive 80% of all infectious mosquito bites in rural African settings, suggesting that malaria parasites may also conform to the "20/80 rule" [3]. Significant malaria risk heterogeneity has also been described in towns and cities in Africa [4–6]. For example, EIRs across the city of Brazzaville were estimated in the early 1980s to range between <1 every three years and >100 per year [7]. Not surprisingly, community-wide EIR measurements are affected by a range of environmental factors (e.g., proximity of houses to water bodies that serve as larval habitats for vectors), behavioural characteristics of individuals (e.g., occupational exposure to mosquitoes and patterns of bednet use), and individual differences in attractiveness to mosquitoes [e.g., 8]. Variation in overall malaria risk may also result from differences in individual susceptibility to infection and subsequent disease given exposure, due to innate resistance and acquired immunity developing after repeated infections [9].

A quantitative understanding of malaria transmission dynamics is required for planning, monitoring, and evaluating interventions aimed at its elimination [10]. However, classical susceptible-infected-susceptible (SIS) malaria models often disregard, totally or partially, risk heterogeneity at the community level and classify hosts as more uniformly susceptible or infectious than they actually are. Models that take insufficient account of real-world heterogeneities may not properly recapitulate the transmission dynamics of malaria in endemic settings, in addition to not providing insights into the impact of targeting control interventions to high-risk groups [1, 10]. SIS models of infectious diseases may incorporate risk heterogeneity among hosts as, for example, a continuous distribution of hosts' susceptibility to infection,

which can be determined empirically from the proportions of hosts that are experimentally infected at different pathogen challenge doses [11–13]. Alternatively, models may assume that the population of susceptible individuals is divided into a finite number of susceptibility classes or frailty groups [13–17].

The incidence of malaria in the Americas has decreased dramatically over the past two decades, but residual transmission pockets persist across the Amazon and challenge current elimination efforts. Residual malaria refers to the transmission that persists despite achieving high coverage of effective control measures such as use of insecticide-treated bednets and indoor residual spraying [18]. *Plasmodium vivax*, the predominant human malaria parasite in the region, is found in nearly 76% of cases in this continent [19]. Here, we fit compartmental SIS models that incorporate risk heterogeneity to malaria surveillance data, aiming to explore the transmission dynamics of *P. vivax* in the main urban malaria hotspot of the Amazon Basin of Brazil.

## Results

### A homogeneous-risk model does not satisfactorily recapitulate the epidemiology of *Plasmodium vivax* malaria

We first fitted empirical data by using a compartmental SIS model that considers the entire host population as being homogeneously at risk ($p_1 = 1$ and $x_1 = 1$; parameters are described in Materials and Methods section) of clinical vivax malaria (Fig 1C). The simultaneous fitting to empirical profiles of incidence by age and number of annual episodes per person (parameter estimation process is fully described in S1 File) is optimal when the age-dependent force of infection (Eq 1) takes parameter values $\lambda_0 = 0.7452$, $c = 0.8787$ and $k = 0.0282$ (Fig 1D) and the partial immunity factor (Eq 2) decays at constant rate $\alpha = 0.1162$ per infection experienced (Fig 1E). The homogeneous-risk model output recapitulates how malaria incidence density varies with age (Fig 1A; see also [20]) but does not satisfactorily fit the number of episodes per person over the one-year follow-up (Fig 1B).

### A 20% fraction of high-risk individuals accounts for 86% of the community-wide malaria burden

We next consider two susceptibility classes (high-risk [HR] and low-risk [LR] groups) to account for risk heterogeneity in the host population. We optimised model fitting (S1 File) for different proportions of individuals in the HR and LR groups, with the best fit corresponding to a model with 20% of the host population allocated to the HR group (Table 1).

The best-fitting solution obtained with the heterogeneous model is presented in Fig 2. Fig 2A compares empirical age-specific malaria incidence data to the model output, which combines incidence densities in the LR and HR groups. Overall, the HR group is estimated to contribute 86.0% of the overall vivax malaria burden in the community, roughly as expected from the "20/80 rule" [1]. High-risk individuals become infected earlier and acquire partial immunity faster than their low-risk counterparts, resulting in markedly different, subgroup-specific age-incidence patterns. In the HR group, the incidence of clinical malaria sharply increases with age among children and adolescents, but declines thereafter; in contrast, malaria incidence density increases slowly in the LR group and reaches a plateau in the fourth decade of life. Fig 2B shows that the model properly fits the empirical frequency distribution of cases per person accumulated over one year of follow-up.

Fig 2C, 2D and 2E show, respectively, the risk distribution, the age-dependent force of infection and the partial immunity factor. The risk distribution has variance $v = 3.3247$ [95%

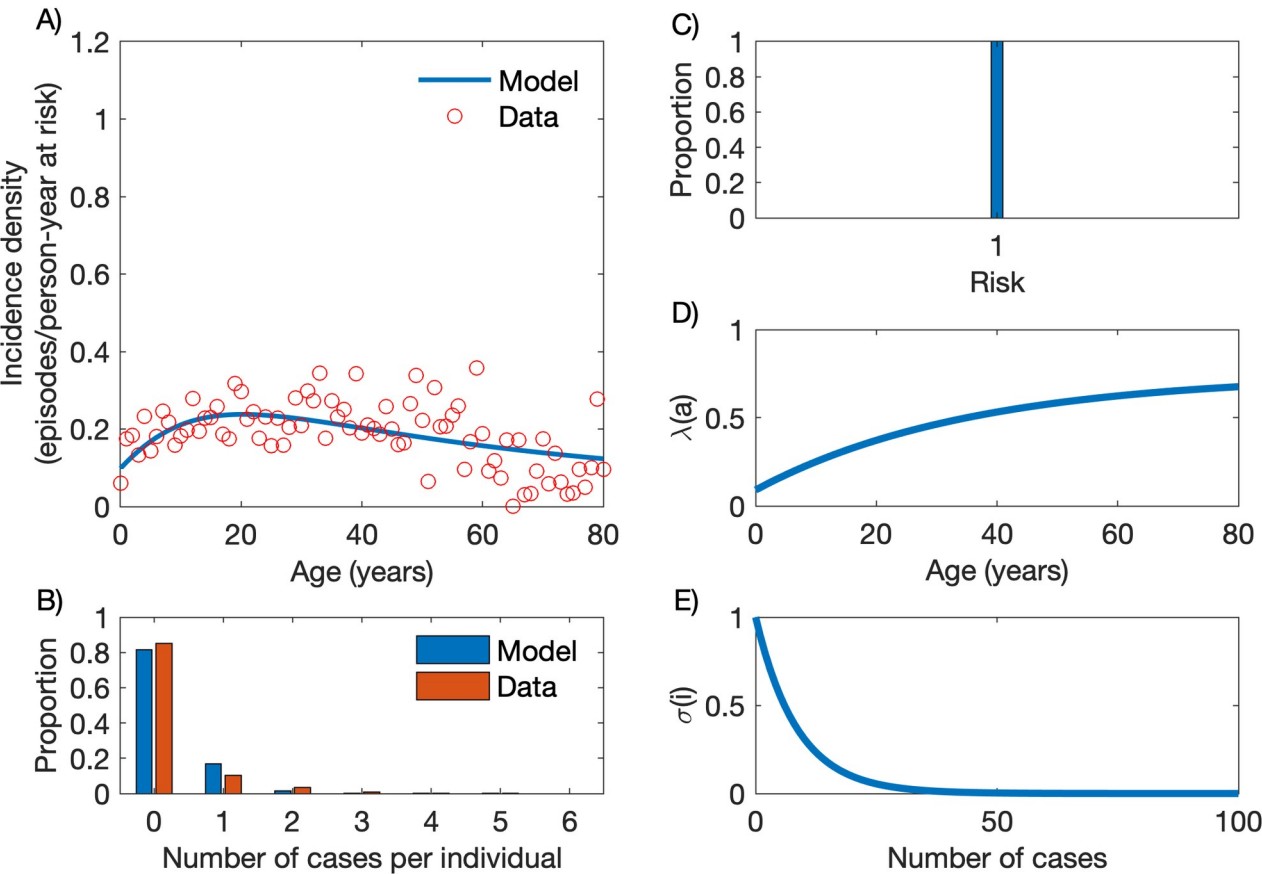

**Fig 1. Model with homogeneous risk.** (A) Age-specific malaria incidence data (red circles) and the best fitting model output (blue line). (B) Frequency distribution of the number of cases per person, empirical data (red bars) and model output (blue bars). (C) Homogeneous risk distribution. (D) Age-dependent force of infection (Eq 1) with parameters $\lambda_0 = 0.7452$, $c = 0.8787$ and $k = 0.0282$. (E) Partial immunity factor (Eq 2) with parameter $\alpha = 0.1162$.

credible interval: 3.1057–3.3845], with 80% ($p_1 = 0.8$) of the population having low risk $x_1 = 0.0883$ [95% CI: 0.0801–0.1189]) (LR group) and 20% ($p_2 = 0.2$) high risk $x_2 = 4.6467$ [95% CI: 4.5246–4.6794]) (HR group). Note that *P. vivax* malaria risk is approximately 26-fold higher in individuals in the $S_{0,2}$ compartment, which comprises malaria-naïve high-risk subjects, compared to their counterparts in the $S_{0,1}$ compartment, which comprises malaria-naïve low-risk subjects. However, this difference changes with age as individuals in each group become infected and acquire partial immunity. The model fits the data optimally when the age-dependent force of infection (Eq 1) takes parameter values $\lambda_0 = 0.6197$ [95% CI: 0.3680–0.7174],

**Table 1. Model fitting for different risk distributions.**

| HR-LR (in %) | Log-likelihood |
|---|---|
| 0–100 | 118.4802 |
| 10–90 | 133.2681 |
| 15–85 | 141.4236 |
| **20–80** | **142.6645** |
| 25–75 | 140.6231 |
| 30–70 | 137.4449 |

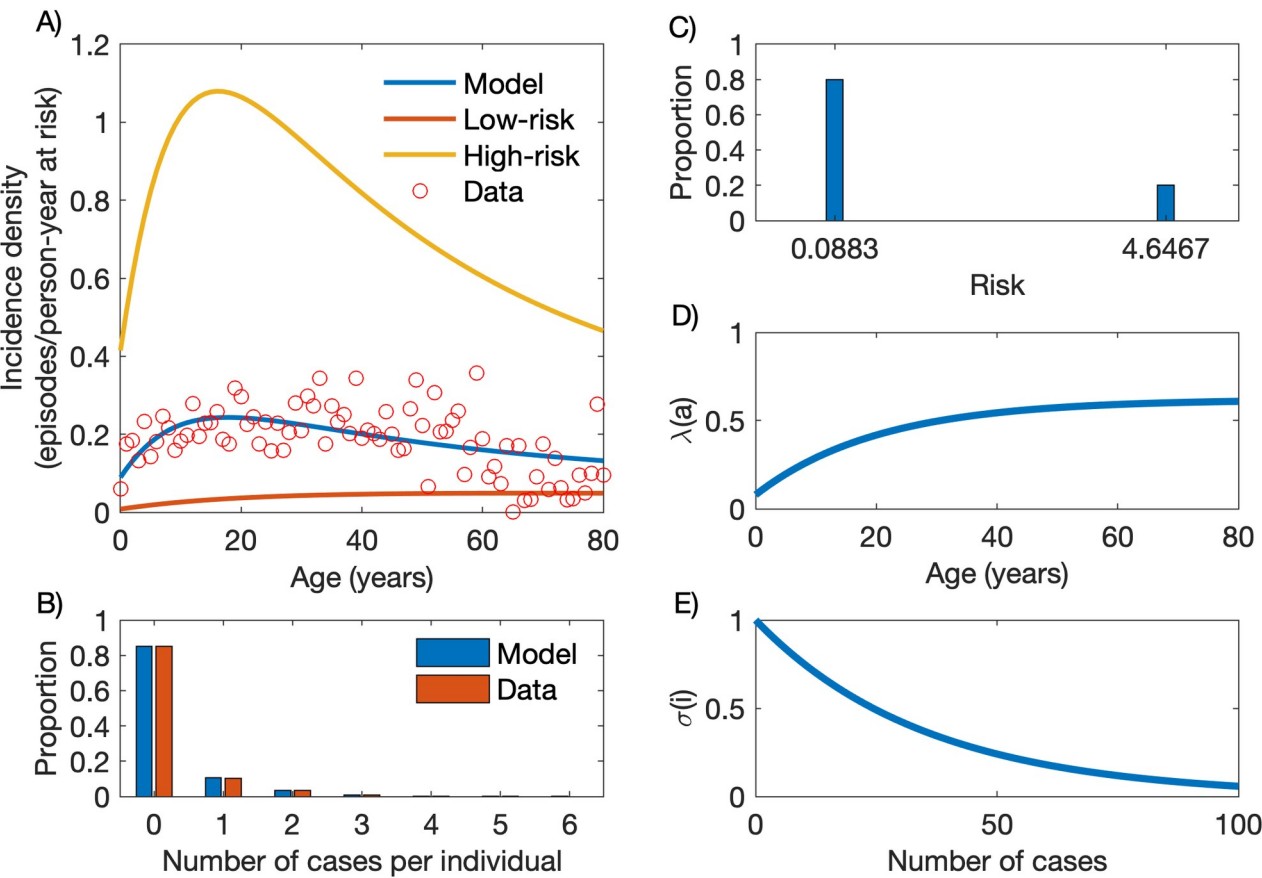

**Fig 2. Model with heterogeneous risk.** (A) Age-stratified incidence data (red circles) and the model output (blue line) as a composition of incidence densities in the low-risk (LR; red line) and high-risk (HR; yellow) groups. (B) Frequency distribution of the number of cases per person, empirical data (red bars) and model output (blue bars). (C) Risk distribution with variance $v = 3.3247$ [95% credible interval: 3.1057–3.3845], partitioning the population into 80% ($p_1 = 0.8$) in the LR group ($x_1 = 0.0883$ [95% CI: 0.0801–0.1189]) and 20% ($p_2 = 0.2$) in the HR group ($x_2 = 4.6467$ [95% CI: 4.5246–4.6794]). (D) Age-dependent force of infection (Eq 1) with parameters $\lambda_0 = 0.6197$ [95% CI: 0.3680–0.7174], $c = 0.8720$ [95% CI: 0.6638–0.9642] and $k = 0.0493$ [95% CI: 0.0392–0.1173]. (E) Partial immunity factor (Eq 2) with parameter $\alpha = 0.0285$ [95% CI: 0.0162–0.0330].

$c = 0.8720$ [95% CI: 0.6638–0.9642] and $k = 0.0493$ [95% CI: 0.0392–0.1173], and the partial immunity factor (Eq 2) decays at rate $\alpha = 0.0285$ per infection [95% CI: 0.0162–0.0330].

## High-risk individuals develop immunity and constitute a clinically silent reservoir of infection

We next incorporate to the model, compartments with individuals who are infected but asymptomatic. The dynamics of individuals through model compartments, considering that asymptomatic infections last an average of 90 days (i.e. $\gamma' = 1/90$ per day), is shown in Fig 3. Individuals in the LR group move slowly between compartments (Fig 3A, 3B and 3C), compared with their HR counterparts (Fig 3D, 3E and 3F). Using the population age structure determined by our census survey, the model predicts that, in the current population, 77.8% and 5.4% of the individuals of the HR and LR groups, respectively, had at least one clinical malaria attack. As a consequence, acquired immunity following repeated *P. vivax* malaria episodes affects almost exclusively the dynamics of HR individuals, leading to frequent asymptomatic infections (Fig 3C and 3F).

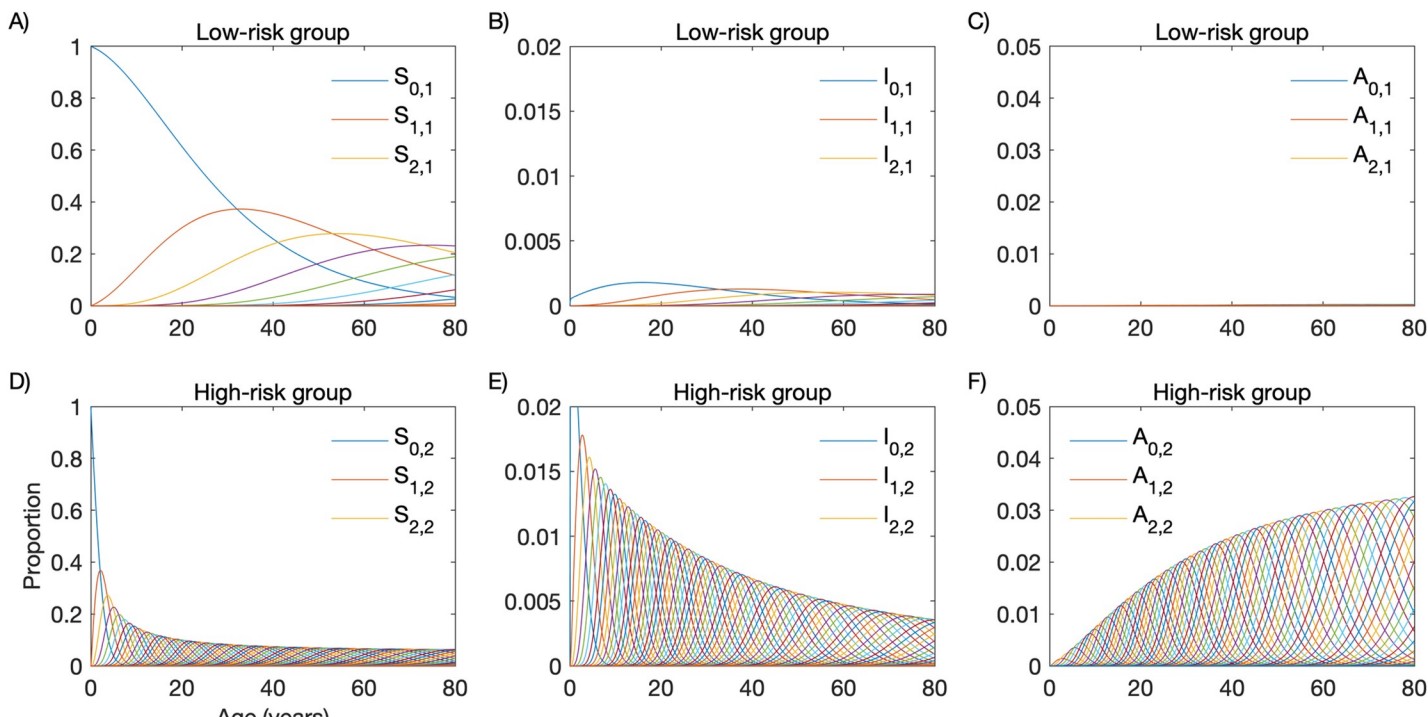

**Fig 3. Age-profiles of repeated malaria in a heterogeneous host population comprising a high-risk (HR) and a low-risk (LR) group.** (A) Susceptible individuals in the LR group; (B) Symptomatic infected individuals in the LR group; (C) Asymptomatic individuals in the LR group; (D) Susceptible individuals in the HR group; (E) Symptomatic infected individuals in the HR group; (F) Asymptomatic individuals in the HR group.

Because the asymptomatic infection recovery rate $\gamma'$ is unknown, we assumed the average duration of asymptomatic parasite carriage ($D_A$) to range from 30 to 180 days (Fig 4). Model outputs recapitulate the age-dependent increase in the prevalence of asymptomatic *P. vivax* carriage that has been described in Amazonian communities (Fig 4A; e.g., [21]) and, as expected, indicate that the community-wide prevalence of asymptomatic *P. vivax* infection increases with longer parasite carriage duration (Fig 4B). Model simulations indicate that HR individuals constitute the vast majority of asymptomatic parasite carriers (Fig 4C). Although this maybe somewhat overrated due to the assumption that acquired immunity reduces symptoms without preventing infection, it highlights plausible trends warranting future empirical studies.

The relative contribution of asymptomatic and symptomatic infections to the overall burden of *P. vivax* infection in the community was also simulated (Fig 5). We observe that, even with short-lived asymptomatic parasite carriage ($D_A = 1/\gamma' = 30$ days) and considering the average duration of symptomatic infections that are diagnosed and treated as either 4, 8, or 12 days, 66–85% of subjects carrying *P. vivax* infection at a given time will be asymptomatic, consistent with empirical estimates from across the Amazon ranging between 52% and 90% [21–24]. We note that these empirical data can be used to estimate $\gamma'$ and $D_A$ in the target populations.

Finally, we simulated the relative contribution of asymptomatic parasite carriers to onwards *P. vivax* transmission in a wide range of plausible scenarios. To this end, we consider that symptomatic and asymptomatic parasite carriers remain infectious for 4, 8 and 12 days and 30, 90 and 180 days, respectively, with a relative infectiousness (*RI*) of asymptomatic compared to symptomatic infections of 1/2, 1/10 and 1/30 (Fig 6). Model outputs indicate that even

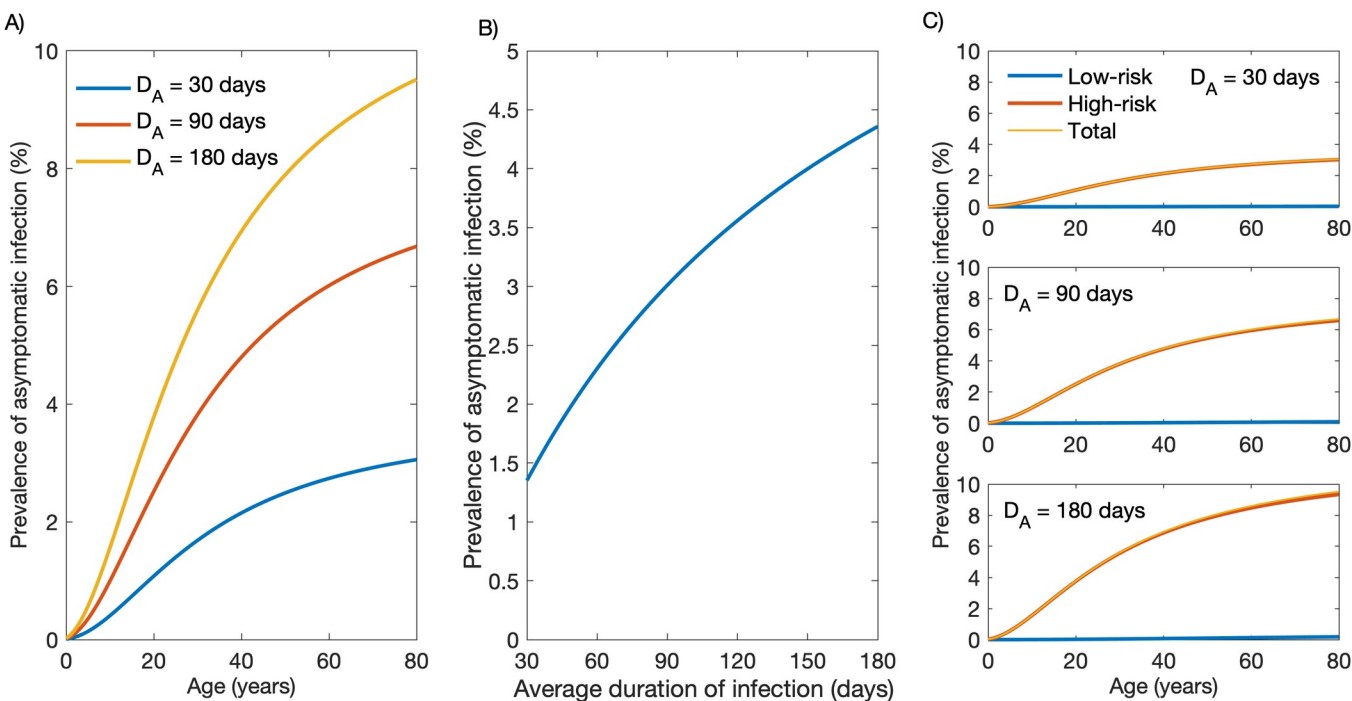

**Fig 4. Prevalence of asymptomatic *Plasmodium vivax* infection according to the average duration of parasite carriage.** (A) Age-stratified prevalence of asymptomatic infection considering an average duration of asymptomatic parasite carriage $D_A$ of 30, 90 and 180 days. (B) Variation in the community-wide prevalence of asymptomatic infection according to the average duration of asymptomatic parasite carriage. (C) Age-stratified prevalence of asymptomatic infection in the low-risk (LR) and high-risk (HR) groups considering an average duration of asymptomatic parasite carriage $D_A$ of 30 days (upper panel), 90 days (middle panel) or 180 days (lower panel).

short-lived asymptomatic *P. vivax* carriage ($D_A$ = 30 days) can contribute substantially to onwards malaria transmission in the community if the overall *RI* ranges between 1/2 and 1/10 (Fig 6A and 6D). Sustained asymptomatic *P. vivax* carriage ($D_A$ = 90 days) can account for 30–87% of the infectious reservoir if *RI* ranges between 1/2 and 1/10 (Fig 6B and 6E), with a minor further increase with $D_A$ = 180 days (Fig 6C and 6F). We further note that, for most $D_A$ and *RI* value combinations, the relative contribution of symptomatic infections to the infectious reservoir can be substantially reduced by providing prompt CQ-PQ treatment to reduce the mean gametocyte clearance time (or average duration of infectiousness) from 12 to 4 days.

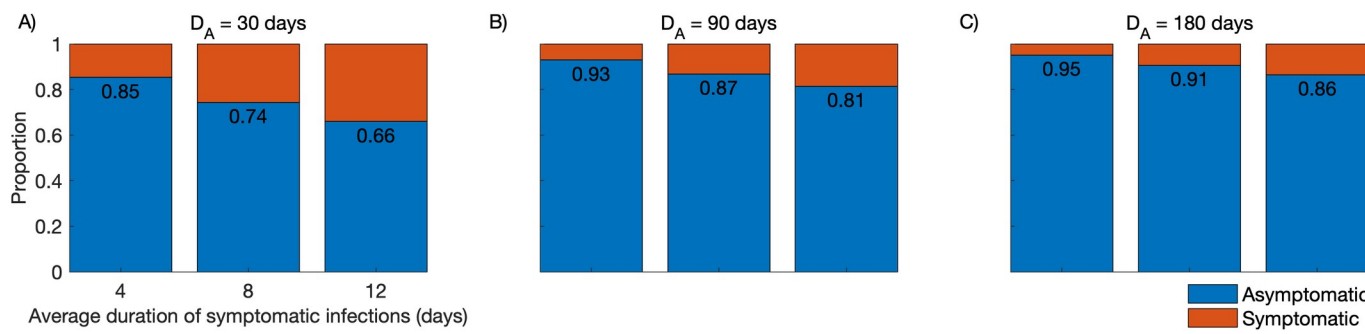

**Fig 5. Simulated proportions of community-wide *Plasmodium vivax* infections that are symptomatic or asymptomatic.** We consider the average duration of symptomatic infections that are diagnosed and treated as either 4, 8, or 12 days; the duration of asymptomatic parasite carriage that remains undetected and untreated ($D_A$) is considered to be 30 days (panel A), 90 days (panel B), or 180 days (panel C).

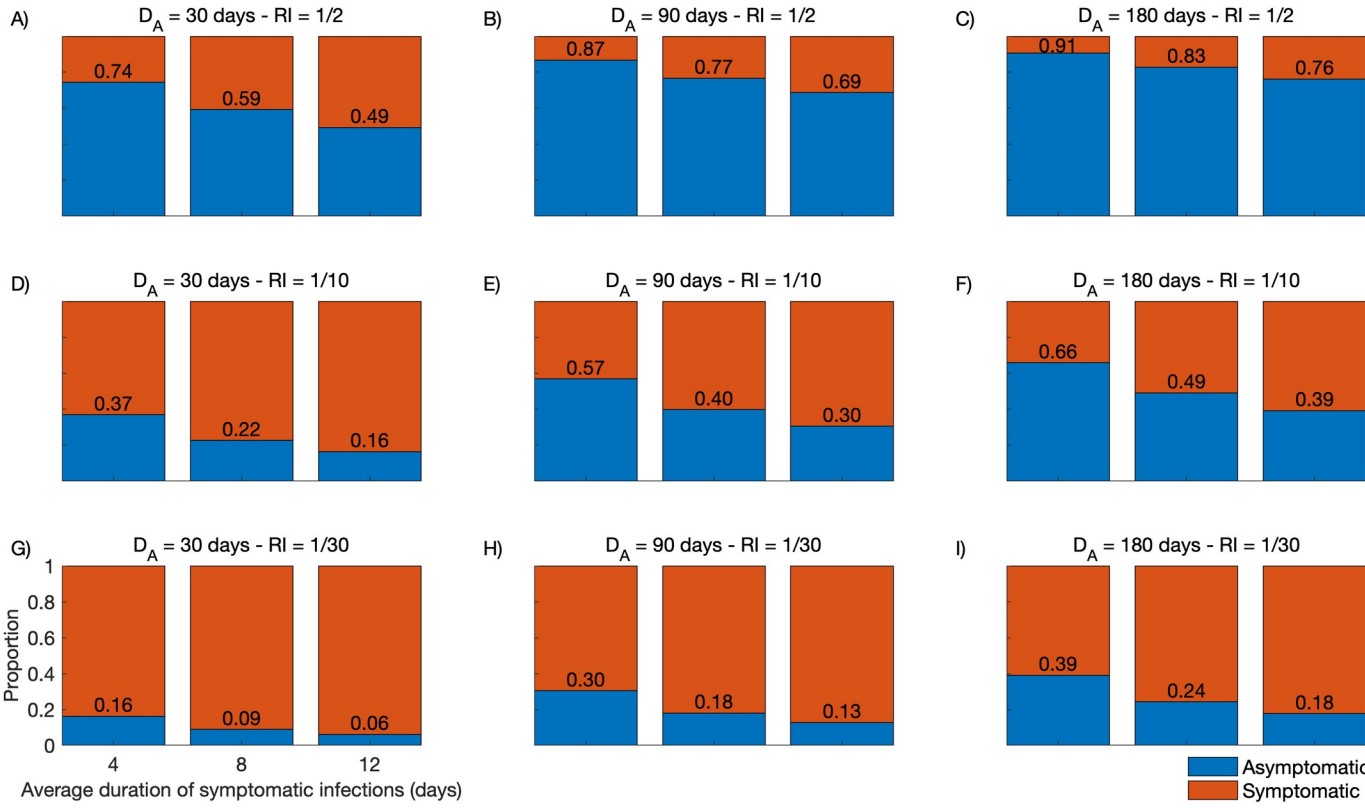

**Fig 6. Relative contribution to the *Plasmodium vivax* infectious reservoir of individuals with symptomatic and asymptomatic infections.** Model outputs consider different average durations of asymptomatic parasite carriage $D_A$ ($D_A$ = 30 days in panels A, D and G; 90 days in panels B, E and H; and 180 days in panels C, F and I) and different relative infectiousness (*RI*) of asymptomatic compared to symptomatic infections (*RI* = 1/2 in panels A, B and C; 1/10 in panels D, E and F; and 1/30 in panels G, H and I). For every combination of $D_A$ and *RI*, we simulated the average duration of infectiousness of symptomatic infections as either 4, 8 or 12 days.

## Discussion

Measuring how malaria infection risk varies among individuals is challenging. Product of exposure to infectious mosquitoes and susceptibility to infection given exposure, each individual's risk is determined by numerous interacting factors. Despite notorious efforts being invested in characterising specific determinants, such as individual mobility to and from hotspots [25], parasite genetics [26] and human genetics [27], a complete catalogue of risk factors and respective measures is not on the horizon. Smith [28] suggested that individual-level variation in susceptibility to malaria given exposure can be inferred by modelling malaria incidence as a function of EIR measured in the same population. Similarly, matched EIR and parasite prevalence data have been used to quantify heterogeneity in malaria susceptibility by assuming a gamma distribution of relative infection rates in the host population [5]. However, the widespread use of these approaches is limited by the restricted availability of reliable EIR measurements, which are notoriously difficult to obtain, from across endemic communities. Malaria transmission models that consider heterogeneity have instead assumed either a small number of measured risk factors or unmeasured ranges of individual risk variation incorporated as either discrete frailty groups or a continuous variable (e.g., [29]).

Here, we show that a compartmental SIS model with heterogeneous risk notoriously outperforms its mean-field approximation in recapitulating the transmission dynamics of *P. vivax* in the main malaria hotspot of Brazil. We provide an empirical basis to estimate risk

heterogeneity in host populations by simultaneously fitting SIS models to two sets of surveillance data—namely, age-related malaria incidence and frequency distribution of malaria cases per person—derived from the same population-based cohort. The best-fitting heterogeneous-risk model considers that the HR group comprises 20% of the host population and contributes 86% of the vivax malaria burden in the community. We suggest that this approach can be used to fit empirical data from across a range of malaria-endemic settings to test whether other host populations conform to this 20/80 pattern.

The estimated force of infection in the main residual malaria hotspot of Brazil is one order of magnitude lower than that estimated for *P. falciparum* in children from across rural Africa (e.g., [20, 30]). As a consequence, our study population appears to acquire partial immunity to malaria rather slowly. Indeed, the model predicts that as much as 25 past clinical malaria attacks, on average, are required in order to reduce by half the risk of a clinical malaria attack. In holoendemic settings, children are typically continuously infected during the transmission season, with frequent superinfection and overlapping clinical malaria episodes during their first years of life. For example, children aged 1–5 years in Papua New Guinea were estimated to experience an average of 2.5 episodes of clinical vivax malaria per year in 2006–2007, before intensified, large-scale control interventions were implemented nationwide [31]. Similarly, in Mali an average of 2.4 episodes of clinical falciparum malaria per child aged 3–59 months per year have been estimated to occur, despite the distribution of long-lasting insecticide-treat bed nets at baseline [32]. Both estimates give an average of 25 malaria attacks by the age of 10–11 years. Indeed, in such areas, malaria remains common throughout most of childhood, and a significant decrease in risk of infection is seen in adolescence and early adulthood. In our study site, although partial immunity develops earlier in the HR group, with a decline in malaria incidence after the second decade of life (Fig 2A), HR individuals across all age groups still constitute the main contributors to the overall clinical malaria burden.

Despite the low overall force of infection in the study area, the fraction of HR individuals who experience repeated *P. vivax* infections and gradually develop partial immunity will eventually become asymptomatic but potentially infectious parasite carriers overlooked by routine surveillance. Although the overall average incidence of clinical *P. vivax* malaria in Mâncio Lima, estimated at 20.90 episodes/100 person-years at risk between October 2015 and September 2016, is substantially lower than that observed in holoendemic settings, some HR individuals may be nearly as exposed to malaria as the average child living in rural Africa. In fact, around one fourth of study subjects experienced one or more episodes of clinical vivax malaria during the study period; 29.9% of those with symptomatic *P. vivax* infections diagnosed during the study period had two or more episodes (Fig 2B, red bars), indicating that a fraction of exposed subjects actually experience repeated *P. vivax* episodes over one year of follow-up. Therefore, the paradoxical finding of clinical immunity and frequent asymptomatic infections in Amazonian communities exposed to low overall levels of malaria transmission [33] can be explained by the presence of a fraction of HR subjects that experience the majority of infections in the community and acquire clinical immunity. Statistical modelling of malaria surveillance data has identified young adult males living in the less urbanized periphery of the town as the main HR individuals in Mâncio Lima [34]. Importantly, these HR individuals not only contribute disproportionately to the overall burden of clinical disease (Fig 2A), but also constitute the silent reservoir of sustained asymptomatic infections (Fig 4C) that are left untreated and may contribute significantly to onwards malaria transmission in this and other low-endemicity settings [35]. Estimates of the proportions of asymptomatic infections that are patent (consistent with *RI* close to 1/2) vary by one order of magnitude, from 4.5% [24] to 46.7% [22], in Amazonian populations.

The importance of characterising malaria reservoirs in endemic regions has recently been highlighted [36] and the results from this work further underscore how essential this

information is to inform elimination programmes for properly planning control interventions. Heterogeneous risk implies that imperfect control measures, such as leaky vaccines, if uniformly applied to the entire host population, are unlikely to reduce substantially the overall malaria burden [29]. Our model simulations, however, suggest that a dramatic reduction in the community-level burden of clinical *P. vivax* malaria can be achieved by selectively targeting HR subjects, if they can be readily identified, to more intensive and effective measures that may not be readily delivered to the entire population.

We have limited the present analysis to *P. vivax*, which predominates in the areas of residual malaria transmission across the Amazon Basin. One major feature of *P. vivax* is that parasites may persist for several months in human hosts as hypnozoites, the dormant liver stages that eventually reactivate and may cause one or more new blood-stage infections termed relapses following a single infectious mosquito bite [37]. Radical cure of vivax malaria thus requires the use of antimalarial drugs that target both blood and liver stages, such as PQ and tafenoquine. Although we do not consider relapses explicitly in our compartmental models, they are implicitly integrated into the force of infection, which combines blood-stage infections arising from infecting stages (sporozoites) inoculated during mosquito bites and relapses arising from reactivating hypnozoites. We hypothesise that HR and LR individuals initially differ in their exposure to infectious mosquitoes or susceptibility to infection and disease once challenged with infecting sporozoites, but over time HR individuals become also more likely to have *P. vivax* relapses originating from the large hypnozoite reservoir that they have accumulated in the liver following repeated infections. Importantly, new infections and relapses entail different control measures; while the incidence of new infections can be reduced by decreasing exposure to mosquito bites, e.g. with insecticide-treated bednets, relapses can be prevented by improved anti-relapse treatments.

The present study has some limitations. First, we used routinely collected malaria morbidity data for model fitting, but blood samples were not available for further confirmatory (e.g., molecular) diagnostic tests. Moreover, surveillance data used to fit our models do not include sub-patent and asymptomatic malaria episodes experienced by the target population. Second, our modelling approach does not allow for estimating the impact of improved anti-relapse therapies on the overall *P. vivax* malaria burden, since we do not differentiate between blood-stage infections arising from hypnozoites and newly inoculated sporozoites. Third, there are no empirical data, obtained in the same population, to properly measure the relative infectiousness of asymptomatic infections, either patent or not, and estimate more precisely their potential contribution to malaria transmission in the community.

We conclude that considering risk heterogeneity in the host population is crucial for properly describing the transmission dynamics of *P. vivax* using compartmental SIS models and provide a framework to test the hypothesis that a few HR subjects contribute the vast majority of the vivax malaria burden at the community level. Moreover, HR subjects are important contributors to the silent infectious reservoir that likely fuels onwards malaria transmission in low-endemicity settings. These results can be further explored for the evidence-based planning and deployment of control interventions towards the elimination of residual P. vivax malaria across the Amazon Basin.

## Materials and methods

### Ethics statement

The study protocol was approved by the Institutional Review Board of the Institute of Biomedical Sciences, University of São Paulo, Brazil (CEPH-ICB 1368/17); written informed consent and assent were obtained for the census survey.

## Study site and population

The study site, the municipality of Mâncio Lima (07˚36' 51"S, 72˚53' 45"W), is situated in the upper Juruá Valley, next to the border between Brazil and Peru. With 17,910 inhabitants (half of them in the urban area) and 9,278 laboratory-confirmed malaria cases notified in 2017, Mâncio Lima has currently the highest annual parasite incidence (API) in Brazil, estimated at 518.0 malaria cases per 1,000 inhabitants. Mâncio Lima is unique in Brazil in that 49% of all local malaria infections are reportedly acquired in urban settings, compared to a country's average of 15% (Ministry of Health of Brazil; SIVEP-Malaria; http://portalweb04.saude.gov.br/sivep_malaria/default.asp; accessed 04 July 2019).

The study cohort comprised 8,783 permanent residents in the town of Mâncio Lima, aged from <1 to 80 years and distributed into 2,329 households. These individuals were systematically enumerated during a demographic census survey carried out by our field team between November 2015 and April 2016. Dates of entry in the study cohort were the subject's date of birth or October 1, 2015, whatever was the most recent; this information was used to calculate the number of person-years at risk for incidence density estimation. For the purposes of this analysis, we assumed that no study participant left the study area before September 30, 2016, when the latest morbidity data were collected.

## Malaria morbidity data

We retrieved all records of laboratory-confirmed clinical malaria cases notified in Mâncio Lima between October 1, 2015, to September 30, 2016. Case records were entered into the electronic malaria notification system of the Ministry of Health of Brazil (SIVEP-Malaria; http://200.214.130.44/sivep_malaria/). Because malaria is a notifiable disease in Brazil and only public health facilities provide laboratory diagnosis and malaria treatment, the electronic malaria notification system is estimated to comprise 99.6% of all clinical malaria cases diagnosed countrywide [38]. However, asymptomatic parasite carriage and persistently subpatent infections, which are not detected by microscopy or commercially available, antigen-based rapid diagnostic tests, may have been overlooked. We used patient's name, gender, and age to link malaria case records to individuals in our census survey database, given the absence of common unique patient identifiers. Name entries were compared using the Jaro-Winkler string distance [39] as implemented in the *stringdist* package of the *R* software [40]. Criteria for associating malaria case records to subjects enumerated during our census survey were: (a) same gender, (b) maximum reported age difference of 1 year, and (c) maximum Jaro-Winkler distance between names of 0.10, with penalty factor of 0.05 (constant scaling factor for how much the score is adjusted downwards for having common prefixes).

A minimal interval of 28 days between two consecutive malaria notifications was required to count the second case as a new malaria episode. When different infecting species were detected in samples obtained less than 28 days apart, the subject was considered to have a single mixed-species infection. Overall, we found 2,057 malaria notifications in the cohort of urban residents during the 12-month study period, with 8,770.8 person-years of follow-up. *P. vivax* accounted for 1,833 cases (89.1%), *P. falciparum* for 193 cases (9.4%) and both species for 31 cases (1.5%). The present analysis is limited to *P. vivax* infections, since this is the most abundant in our study location. Describing the transmission dynamics of multiple *Plasmodium* species would escalate model complexity and assumptions beyond the realm of the current study. We found an average malaria vivax incidence density of 20.90 episodes/100 person-years at risk. By combining demographic information and malaria morbidity data, we computed age-specific vivax malaria incidence densities and the number of vivax malaria episodes per person recorded in the urban cohort over 12 months. These empirical data were used to fit model outputs.

**Fig 7. Susceptible-infected-susceptible (SIS) compartmental model representing the dynamics of malaria over age in a heterogeneous host population.** The compartments describe the following epidemiological classes: $S_{i,j}$ represents susceptible individuals from risk group $j$ (1 = low-risk [LR]; 2 = high-risk [HR]) who have experienced $i$ past clinical malaria attacks; $I_{i,j}$ represents symptomatic infected individuals from risk group $j$ who are currently experiencing their $i$th clinical malaria attack. Individuals experience new infections due to an age-dependent force of infection $\lambda(a)$ modified by a risk factor $x_j$, and a partial immunity weight $\sigma(i)$; all individuals recover at the same rate $\gamma$.

## The mathematical model

The compartmental SIS model describing the epidemiology of clinical vivax malaria is represented diagrammatically in Fig 7. Any population of susceptible individuals is heterogeneous with regards to risk of infection. Individual risk is a continuous characteristic which we discretise in two groups: low risk (LR) and high risk (HR). This is a coarse description of individual heterogeneity that nevertheless suffices to our modelling purposes of capturing the effects of variance in risk. Within each risk group, individuals are classified as either susceptible or infected. Each risk group comprises a proportion $p_j$ ($0<p_j<1$, $j = 1,2$ and $p_1+p_2 = 1$) of the total population and is associated with a risk factor $x_j>0$ ($j = 1,2$). Without loss of generality, we assume that the overall average risk is equal to 1 ($x_1p_1+x_2p_2 = 1$) since the factors $x_j$ are modifiers of individual responses to a force of infection which will be allowed to vary. This setting configures a risk distribution with variance $v = p_1(x_1-1)^2+p_2(x_2-1)^2$.

We assume an age-dependent force of infection $\lambda(a)$ (Eq 1), which correlates mosquito biting activity with human body mass [30, 41]. This function strictly increases with age, with minimum $\lambda_0(1-c)$ (at age zero) and upper limit $\lambda_0$. The parameter $k$ determines how steeply the force of infection increases in early ages and $c$ controls the value at birth.

$$\lambda(a) = \lambda_0(1 - ce^{-ka}) \tag{1}$$

Assuming that individuals acquire partial immunity after repeated clinical malaria attacks, due to antibody- and cell-mediated responses [42], we introduce a factor describing the development of partial immunity. The strictly positive decreasing function $\sigma(i)$ of the number $i$ ($i \geq 0$) of past clinical vivax malaria attacks each individual has experienced (Eq 2), with a maximum for malaria-naïve individuals ($\sigma(0) = 1$), simulates a partial immunity factor and weights down the age-dependent force of infection $\lambda(a)$ as the number of cumulative clinical malaria episodes increases. The factor describing partial immunity is controlled by the parameter $\alpha$, which determines the rate at which immunity develops after repeated infections.

$$\sigma(i) = e^{-\alpha \cdot i} \tag{2}$$

Assuming equilibrium with respect to time, in addition to the age-dependent force of infection, partial immunity acquisition and risk heterogeneity, malaria unfolds in age domain according to a system of ordinary differential equations (ODEs) (system of Eq 3), with initial

conditions $S_{0,j}(0) = p_j$, $S_{i,j}(0) = I_{i,j}(0) = 0$, for $i = 1,2,\dots$ and $j = 1,2$.

$$\frac{dS_{0,j}}{da} = -x_j\sigma(0)\lambda(a)S_{0,j}$$

$$\frac{dI_{1,j}}{da} = +x_j\sigma(0)\lambda(a)S_{0,j} - \gamma I_{1,j}$$

$$\frac{dS_{1,j}}{da} = -x_j\sigma(1)\lambda(a)S_{1,j} + \gamma I_{1,j}$$

$$\frac{dI_{2,j}}{da} = +x_j\sigma(1)\lambda(a)S_{1,j} - \gamma I_{2,j}$$

$$\vdots$$

$$\frac{dS_{n-1,j}}{da} = -x_j\sigma(n\text{-}1)\lambda(a)S_{n-1,j} + \gamma I_{n-1,j}$$

$$\frac{dI_{n,j}}{da} = +x_j\sigma(n\text{-}1)\lambda(a)S_{n-1,j} - \gamma I_{n,j}$$

$$\vdots$$

(3)

Individuals in the LR group are initially allocated to compartment $S_{0,1}$, comprising susceptible individuals who are malaria-naïve. At a rate which is determined by the age-dependent force of infection $\lambda(a)$ and the risk factor $x_1$, LR individuals move to compartment $I_{1,1}$ after experiencing their first clinical vivax malaria attack. After recovering (with recovery rate $\gamma$), they become susceptible again and move to the next compartment $S_{1,1}$, which comprises susceptible individuals who have already experienced one past malaria attack and acquired some degree of partial immunity. These individuals may acquire a second infection, according to the same age-dependent force of infection and risk factor, but now weighted down by the partial immunity $\sigma(1)$. LR individuals can move forward between compartments within the LR group. With similar dynamics, HR individuals move forward within the HR group, but with a risk factor $x_2$ (Fig 7). This is denominated as the heterogeneous risk model.

For comparison purposes, we built a similar compartmental model where the same average risk is applied to the entire host population (homogeneous risk model; $p_1 = 1$ and $x_1 = 1$, e.g., [20]). We fitted the heterogeneous and the homogeneous risk models to empirical data and compared their ability to recapitulate the epidemiology of vivax malaria in the study population.

## Mathematical model with asymptomatic infections

We refined the SIS model with compartments comprising infected but asymptomatic individuals, by assuming that the proportion of asymptomatic infections depends on gradually acquired partial immunity. This partial immunity is sometimes termed "clinical" or "anti-disease immunity" to emphasise that individuals remain susceptible to infection but become gradually less likely to develop clinical disease once infected. We followed the same basic assumptions of the first model: susceptible individuals from risk group $j$, with age $a$ and with $i$ past clinical attacks ($S_{i,j}(a)$) develop their $i$th clinical case at rate $x_j\sigma(i)\lambda(a)$. Partial immunity developed after $i$ past attacks (Eq 2) reduces by $1-\sigma(i)$ the probability of susceptible individuals $S_{i,j}(a)$ presenting clinical symptoms once infected again. Note that in this model rates of clinical malaria episodes decline explicitly due to clinical immunity, in contrast with the previous implementation which did not specify whether these declines were due to immunity against disease or against infection. Infected subjects thus move to the asymptomatic compartment $A$

**Fig 8. Susceptible-infected-susceptible (SIS) compartmental model representing the dynamics of malaria in a heterogeneous host population considering asymptomatic infections.** The compartments correspond to the following epidemiological classes: $S_{i,j}$ represents susceptible individuals from risk group $j$ (1 = low-risk [LR]; 2 = high-risk [HR]) who have experienced $i$ clinical malaria attacks; $I_{i,j}$ represents individuals with symptomatic infection from risk group $j$ who are currently experiencing their $i$th clinical malaria attack; $A_{i,j}$ represents individuals with asymptomatic infections from risk group $j$ with $i$ past clinical malaria attacks. Individuals experience malaria episodes due to an age-dependent force of infection $\lambda(a)$ modified by a risk factor $x_j$, and a partial immunity weight $\sigma(i)$. Individuals from compartments $I$ and $A$ recover and become susceptible again at rates $\gamma$ and $\gamma'$, respectively.

if they do not develop clinical malaria upon infection. More formally, susceptible individuals $S_{i,j}(a)$ become infected but asymptomatic $A_{i,j}(a)$ at rate $x_j(1-\sigma(i))\lambda(a)$. Individuals with asymptomatic infections from group $j$, age $a$ and who experienced $i$ past clinical malaria attacks ($A_{i,j}(a)$) can eventually progress to their $i$th new clinical attack, at rate $x_j\sigma(i)\lambda(a)$, or recover and become susceptible again at rate $\gamma'$. The compartmental SIS model considering asymptomatic infections is represented diagrammatically in Fig 8.

We assume that naïve individuals from compartment $S_{0,j}(a)$ cannot remain asymptomatic once infected for the first time, since they have not yet developed partial immunity. Indeed, with acquired immunity modelled by an exponential function (Eq 2), we have for naïve individuals $\sigma(0) = 1$. Therefore, the probability of naïve individuals becoming infected but asymptomatic is 0 ($x_j.0.\lambda(a)$).

Introducing asymptomatic compartments ($A$) to the model does not change the dynamics of symptomatic infections, which are represented by our empirical morbidity data. With the assumptions described above, both susceptible and infected but asymptomatic individuals are at risk of symptomatic infection; therefore, the incidence of clinical malaria and the frequency distribution of clinical cases per person remain the same for both models. We thus apply the same parameters estimated in the first model (parameter estimation process is fully described in S1 File), but can now distinguish uninfected and susceptible individuals from those who are infected but remain asymptomatic, according to the recovery rate $\gamma'$.

## Asymptomatic parasite carriers, duration of infection and the infectious reservoir

We simulated several scenarios to address the relative contribution of asymptomatic parasite carriers to the overall burden of infection and onwards transmission in the community. First, we assume individuals with asymptomatic infections to be 2, 10 and 30 times less infectious to mosquitoes than individuals with symptomatic infections (relative infectiousness [RI] of 1/2, 1/10 and 1/30, respectively). Empirical RI estimates vary widely according to the average gametocyte density [43] and are close to 1/2 for microscopy-detected asymptomatic *P. vivax* infections in Ethiopia [44] but range from 1/14 to 1/29 for asymptomatic infections in Colombia and Brazil that can be detected only by molecular methods [45, 46].

Next, we assume that, on average, symptomatic infections can be detected by laboratory methods during 4, 8 and 12 days. Symptomatic infections are curtailed by treatment and their

average length primarily depends on: (a) the duration of the patent but subclinical period that precedes full-blown disease manifestations, which remains elusive; (b) the mean time from the appearance of symptoms to the introduction of chloroquine (CQ)-primaquine (PQ) treatment (2.7 days in our population [47]), and (c) the mean *P. vivax* clearance time following CQ-PQ treatment (1.9 day in our population; [47]). We thus divided the proportion of individuals within the infected (*I*) compartments by 7 (= 28/4), 3.5 (28/8) or 2.3 (28/12) to represent the prevalence of symptomatic blood-stage infections that can be detected by laboratory methods during the subject's 28-day stay in the *I* compartments.

We further assume that asymptomatic blood-stage infections undetected by routine surveillance and left untreated can last between 30 and 180 days. Empirical evidence is rather limited in this area and the duration is clearly context-specific. Once detected by microscopy, asymptomatic *P. vivax* infections in 4 years-old Papua New Guinean children lasted on average 15 days [48], but the time elapsed before blood-stage parasite detection has not been measured. If asymptomatic *P. vivax* infections were first sampled at a random time point during their trajectory, the time to parasite clearance after detection (15 days) is expected to equal, on average, the time elapsed before parasite detection, giving a total duration of 30 days. Here we simulate scenarios with asymptomatic *P. vivax* infections between 30 and 180 days, which corresponds to the median duration of asymptomatic *P. vivax* infections in a cohort study in Vietnam [49].

Finally, we consider the duration of infectiousness to equal the total duration of blood-stage infection in both symptomatic and asymptomatic carriers, under the assumption that virtually all blood-stage *P. vivax* infections comprise mature gametocytes [22,50]. Empirical data from Brazil show that vivax malaria patients become little infectious within 10 hours of CQ-PQ treatment [51], but untreated asymptomatic carriers of subpatent *P. vivax* parasitemia may remain infectious for up to 2 months after parasite detection [52].

## Supporting information

**S1 File. Parameter estimation process.**
(PDF)

## Acknowledgments

We thank Anaclara Pincelli, Odaílton A. Nery and Dr. Nathália F. Lima for carrying out field work, Maria José Menezes for administrative support and Dr. Jan Hasenauer and Elba Raimundez (Institute of Computational Biology, Helmholtz Zentrum München, München, Germany) for helpful discussions.

## Author Contributions

**Conceptualization:** Rodrigo M. Corder, Marcelo U. Ferreira, M. Gabriela M. Gomes.

**Data curation:** Rodrigo M. Corder, Marcelo U. Ferreira.

**Formal analysis:** Rodrigo M. Corder, Marcelo U. Ferreira, M. Gabriela M. Gomes.

**Funding acquisition:** Rodrigo M. Corder, Marcelo U. Ferreira, M. Gabriela M. Gomes.

**Investigation:** Rodrigo M. Corder, Marcelo U. Ferreira, M. Gabriela M. Gomes.

**Methodology:** Rodrigo M. Corder, Marcelo U. Ferreira, M. Gabriela M. Gomes.

**Project administration:** Marcelo U. Ferreira, M. Gabriela M. Gomes.

**Resources:** Marcelo U. Ferreira, M. Gabriela M. Gomes.

**Software:** Rodrigo M. Corder.

**Supervision:** Marcelo U. Ferreira, M. Gabriela M. Gomes.

**Validation:** Rodrigo M. Corder.

**Visualization:** Rodrigo M. Corder, Marcelo U. Ferreira, M. Gabriela M. Gomes.

**Writing – original draft:** Rodrigo M. Corder, Marcelo U. Ferreira, M. Gabriela M. Gomes.

**Writing – review & editing:** Rodrigo M. Corder, Marcelo U. Ferreira, M. Gabriela M. Gomes.

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
