## [Decision Letter · Decision Letter 0]

27 Oct 2019

Dear Dr Corder,

Thank you very much for submitting your manuscript 'Modelling the epidemiology of residual Plasmodium vivax malaria in a heterogeneous host population: a case study in the Amazon Basin' for review by PLOS Computational Biology. Your manuscript has been fully evaluated by the PLOS Computational Biology editorial team and in this case also by independent peer reviewers. The reviewers appreciated the attention to an important problem, but raised some substantial concerns about the manuscript as it currently stands. While your manuscript cannot be accepted in its present form, we are willing to consider a revised version in which the issues raised by the reviewers have been adequately addressed. We cannot, of course, promise publication at that time.

Sincerely,

Jennifer A. Flegg

Guest Editor

PLOS Computational Biology

Rob De Boer

Deputy Editor

PLOS Computational Biology

[LINK]

Reviewer's Responses to Questions

**Comments to the Authors:**

Reviewer #1: Manuscript Number: PCOMPBIOL-D-19-01472

Thank you for inviting me to review the manuscript titled “Modelling the epidemiology of residual Plasmodium vivax malaria in a heterogeneous host population: a case study in the Amazon Basin ” by Corder et al.

The manuscript used mathematical models to estimate the risk of malaria in the Amazon region while accounting for geographical heterogeneity of the study area.

I have the following comments for major revision.

Major comments:

1. At times the introduction blurred into the results, and the results blurred into the discussion.

For example,

(i) Line 82-89 should be part of the results and not introduction.

(ii) Line 460-464 should be moved to the discussion. The results section should only consist of results. May I suggest you align and insert these sentences into the first paragraph of the discussion.

2. The methodology (mathematical models) is too long consisting about 8 pages of different techniques. Could the authors summarize the techniques into a maximum of 2 pages and moved the detailed mathematical models to supplementary materials.

Minor comments

3. Line 52-53: Insert some references here.

4. The authors mentioned in line 52-53 that varying malaria risks have been observed in several towns and cities in Africa countries. However, the example provided was in the city of Brazzaville (Not an African city/country).

5. Line 54-58: Insert references here.

6. Line 77-78: It will be nice to have a sentence describing what residual transmission is.

7. Line 216: …the antimalaria drugs used for radical cure of vivax malaria in Brazil. Provide a reference for this statement.

8. Line 377. Here and throughout the manuscript insert “95% CI” in front of the 95% Credible Interval (CI). Eg. 0.0883 [95% CI: 0.0801-0.1189]

9. Line 506: “…the model predicts that as much as 25 past malaria…” By 25 in this statement, are they saying 25% of past malaria?

Reviewer #2: This manuscript presents fine work. It is based on very good data: a town in Brazil with high malaria incidence, monitored during one year, person by person. This is a piece of exceptional data.

The work adresses the question of risk heterogeneity, an important one. And it proposes a mathematical model that is shown to fit very well the data.

So far, so good, but I would like to raise some points.

1- I don’t really get the point why the two risk classes do not interact. Are the people in those two risk classes somehow geographically separated? Because, if they are mixed, wouldn’t it be the case of a HR person being a potential spreader in the LR population. Please clarify the assumptions of the model at this point.

2-Could you make explicit the parameters that are being fitted. I understand that including age classes is realistic, but also introduces new parameters. Expression (1), for instance has two parameters. Lambda zero is not a problem, but the parameter “k”in Eq.(1) seems arbitrary. Is it a free parameter to be fitted?

3- The model predicts that “As much as 25 past malaria attacks are required in order to reduce by half the risk of a clinical malaria attack”. Is this reasonable? Any other studies show this ? It would be nice to see a discussion on this point.

As I mentioned above, this is a good work, and deserves to be published in a good journal. But it would be interesting to have a more thorough discussion on the model assumptions and parameter “proliferation” .

**Have all data underlying the figures and results presented in the manuscript been provided?**

Reviewer #1: Yes

Reviewer #2: Yes

PLOS authors have the option to publish the peer review history of their article (what does this mean?). If published, this will include your full peer review and any attached files.

Reviewer #1: No

Reviewer #2: No

---

## [Decision Letter · Decision Letter 1]

13 Dec 2019

Dear Dr Corder,

Thank you very much for submitting your manuscript, 'Modelling the epidemiology of residual Plasmodium vivax malaria in a heterogeneous host population: a case study in the Amazon Basin', to PLOS Computational Biology. As with all papers submitted to the journal, yours was fully evaluated by the PLOS Computational Biology editorial team, and in this case, by independent peer reviewers. The reviewers appreciated the attention to an important topic but identified some aspects of the manuscript that should be improved.

We would therefore like to ask you to modify the manuscript according to the review recommendations before we can consider your manuscript for acceptance. Your revisions should address the specific points made by each reviewer and we encourage you to respond to particular issues Please note while forming your response, if your article is accepted, you may have the opportunity to make the peer review history publicly available. The record will include editor decision letters (with reviews) and your responses to reviewer comments. If eligible, we will contact you to opt in or out.raised.

- Supporting Information uploaded as separate files, titled 'Dataset', 'Figure', 'Table', 'Text', 'Protocol', 'Audio', or 'Video'.

We hope to receive your revised manuscript within the next 30 days. If you anticipate any delay in its return, we ask that you let us know the expected resubmission date by email at ploscompbiol@plos.org.

Sincerely,

Jennifer A. Flegg

Guest Editor

PLOS Computational Biology

Rob De Boer

Deputy Editor

PLOS Computational Biology

[LINK]

Reviewer's Responses to Questions

**Comments to the Authors:**

Reviewer #2: As mentioned in my first review, I consider that this manuscript presents fine work. My questions have been partially answered.

Regarding the following statement of the authors:

"Indeed, the model predicts that as much as 25 past clinical malaria attacks are required in order to reduce by half the risk of a clinical malaria attack. In holoendemic settings, children are typically continuously infected during the transmission season, with frequent superinfection and overlapping clinical malaria episodes during their first years of life. "

Could the authors provide a reference. This is a strong and quantitative (25 cases) statement and it just floats around.

A reference for "In other words, HR individuals in our Amazonian study population are nearly as

545 exposed to malaria as the average child living in rural Africa." would also be welcome.

These are minor points that can be easily assessed.

------------

As an aside: moving all equations and mathematical details to S1 resulted in a poorer text. Equations speak by themselves and are much clearer than a description in words. Anyhow, this was done because the other reviewer ask it, so I will not suggest to go back to the previous version.

**Have all data underlying the figures and results presented in the manuscript been provided?**

Reviewer #2: Yes

PLOS authors have the option to publish the peer review history of their article (what does this mean?). If published, this will include your full peer review and any attached files.

Reviewer #2: No

---

## [Decision Letter · Decision Letter 2]

29 Jan 2020

Dear Mr. Corder,

We are pleased to inform you that your manuscript 'Modelling the epidemiology of residual Plasmodium vivax malaria in a heterogeneous host population: a case study in the Amazon Basin' has been provisionally accepted for publication in PLOS Computational Biology.

Before your manuscript can be formally accepted you will need to complete some formatting changes, which you will receive in a follow up email. A member of our team will be in touch within two working days with a set of requests.

Best regards,

Jennifer A. Flegg

Guest Editor

PLOS Computational Biology

Rob De Boer

Deputy Editor

PLOS Computational Biology

Reviewer's Responses to Questions

**Comments to the Authors:**

Reviewer #2: I have no further comments, and I think that now the paper is ready for publication.

**Have all data underlying the figures and results presented in the manuscript been provided?**

Reviewer #2: Yes

PLOS authors have the option to publish the peer review history of their article (what does this mean?). If published, this will include your full peer review and any attached files.

Reviewer #2: No